# Aging and Oral Care: An Observational Study of Characteristics and Prevalence of Oral Diseases in an Italian Cohort

**DOI:** 10.3390/ijerph16193763

**Published:** 2019-10-07

**Authors:** Dorina Lauritano, Giulia Moreo, Francesco Carinci, Raffaele Borgia, Alberta Lucchese, Maria Contaldo, Fedora Della Vella, Patrizia Bernardelli, Guido Moreo, Massimo Petruzzi

**Affiliations:** 1Department of Medicine and Surgery, Centre of Neuroscience of Milan, University of Milano-Bicocca, 20126 Milan, Italy; moreo.giulia@gmail.com (G.M.); raffaeleborgia@yahoo.it (R.B.); 2Department of Morphology, Surgery and Experimental medicine, University of Ferrara, 44121 Ferrara, Italy; crc@unife.it; 3Multidisciplinary Department of Medical-Surgical and Dental Specialties, University of Campania—Luigi Vanvitelli, 80138 Naples, Italy; alberta.lucchese@unicampania.it (A.L.); maria.contaldo@gmail.com (M.C.); 4Interdisciplinary Department of Medicine, University of Bari, 70121 Bari, Italy; dellavellaf@gmail.com (F.D.V.); massimo.petruzzi@uniba.it (M.P.); 5San Carlo-RSA Bernardelli Hospital, 20037 Paderno Dugnano (MI), Italy; bernardelli@clinicasancarlo.it; 6San Carlo Hospital, Internal Medicine Department, 20037 Paderno Dugnano (MI), Italy; guido.moreo@clinicasancarlo.it

**Keywords:** oral health, edentulism, dementia, geriatric (aging), elderly patients

## Abstract

*Background*: Poor oral health is a common condition in patients suffering from dementia. Several aspects of this systemic pathology contribute to causing oral problems: cognitive impairment, behavior disorders, communication and, motor skills deterioration, low levels of cooperation and medical-nursing staff incompetency in the dental field. *Objectives*: The objectives of this study were to evaluate the prevalence and the characteristics of oral pathology in a demented elderly population, as well as to check the association between the different degree of dementia and the oral health condition of each patient. *Materials and Methods*: In this observational study (with cross-sectional design) two groups of elderly patients suffering from dementia, living in two different residential care institutions were recruited. The diagnosis of dementia of each included patient was performed using the Clinical Dementia Rating Scale. In order to evaluate the oral health condition of the included subjects, each patient underwent a physical examination of the oral cavity, during which different clinical parameters were analyzed (number of remaining teeth, oral mucosa, periodontal tissues, bone crests). To each parameter, a score was assigned. Spearman’s Rho test was used. *Results*: Regarding the prevalence of oral pathology in elderly suffering from dementia, it emerged that 20.58% of the included patients had mucosal lesions and/or new mucosal formations (in most cases undiagnosed and therefore untreated). The prevalence of periodontal disease was equal to 82.35% and a marked clinically detectable reabsorption of bone crests was found in almost all patients (88.23%). 24.13% of patients, who underwent the oral examination, had totally edentulous maxillae and/or with retained roots, without prosthetic rehabilitations. The correlation index r showed the presence of a linear correlation (inverse relationship) between the degree of dementia and the state of health of the oral cavity of each patient. *Conclusions*: Several factors contribute to poor oral health in the elderly suffering from dementia: cognitive functions deterioration, behavioral disorders and inadequate medical-staff nursing training on oral hygiene. This study also demonstrated that the lower the dementia degree is, the lower tends to be the oral health status. In order to guarantee a complete assistance to these patients, residential care institutions should include in their healthcare program specific dental protocols.

## 1. Background

Oral health is an essential component of general systemic health: it contributes to physical and mental well-being. To ensure each patient the most appropriate care, the dentist’s practice should not be exclusively focused on the teeth and supporting structures, but it should be extended to cover the oral health needs of patients in a holistic approach. Besides the treatment of specific oral pathologies, the dentist of the future should have the ability to diagnose and treat oral mucosal lesions, orofacial traumas, infections, pain symptoms, manifestations of systemic, genetic and congenital diseases [1]. Dentists should not only be perfectly updated on techniques and materials, but should also know how dental treatments can affect the individual’s state of health/disease and how, at best, dental care should be conducted in subjects suffering from systemic diseases. In fact, medical and surgical conditions could influence oral health and oral healthcare, which could conversely have an impact on general health and healthcare [2]. Le Bars et al. [3] confirmed this theory, reviewing the dental literature concerning the impact of the removable prosthesis on the health status of medically compromised patients. The author showed that patient with dentures, suffering from certain systemic disease (diabetes mellitus, cancer, cardiovascular disease etc.), has higher levels of the organism *Candida albicans* in the oral flora compared to healthy patients. As a recent prospective study underlined, the age of the patients, the presence of one or more chronic disorders and the patients’ drug regimens can influence dental treatment and oral health [4]. For this reason, the treatment approach in dentistry should always include medical history, in order to obtain the greatest possible number of information on the patient’s state of health. Collecting medical history has the objective of defining whether or not it is appropriate to proceed with a therapy, to verify whether the patient is able to sustain the treatment and if some drugs are contraindicated [5]. Jainkittivong et al. highlighted that medical consultations could reduce the medical risk associated with dental procedures and unnecessary antibiotic prophylaxis. Therefore, for many dental patients, a good communication between dentists and physicians is essential for inappropriate care [6]. Unfortunately, the importance of medical history in oral care is often underestimated and it is consequently often neglected [4].

Among systemic pathologies an important role is covered by dementia, a mental condition, characterized by the evidence of cognitive decline in one or more domains (memory, language, executive functions, social cognition etc.), which have an impact on functional autonomy of the subject [7]. One of the most important risk factors associated with the onset of dementia is age and, in an aging society, the impact of the phenomenon is projected to become overwhelming: it is easy to foresee that these pathologies will become, in a short time, one of the most important problems in terms of public health [8]. In support of this, Rizzi et al. showed that dementia prevalence increases from 2–3% among people aged 70–75 years to 20–25% among elderly aged 85 years or more [9]. Studies included in two recent systematic reviews by Delwel et al. suggested that poor oral health is a common condition among the elderly with dementia. Compared to non-demented elderly, people with dementia have a higher prevalence of periodontal disease, mucosal lesions, caries, plaque, reduced salivary flow [10,11], outlining an association between poor oral health and dementia [12]. Functional limitations associated with impairment in different cognitive domains, such as, difficulty in holding information in mind, in organizing or planning, inability in following directions, loss of judgment and inappropriate-aggressive behavior could explain the higher prevalence of oral health problems in dementia subjects [13]. The demented may have problems in communicating, remembering, and performing the simplest actions. In the initial stage the most common symptom is the short-term memory failure; the patient can forget a specific word and use elaborate peripheral compensations; even the daily activities previously mastered (money management, driving a car, housekeeping) can become difficult. Personality changes, emotional frailty is possible. As patients progress to the intermediate stage of dementia, their ability to perform basic daily activities (washing, dressing) is reduced. Patients with severe dementia are no longer able to perform daily activities and become totally dependent on others for food, personal hygiene and movements. Due to cognitive impairment elderly with dementia are no longer able to take care of their oral hygiene, they rarely manage to communicate pain or discomfort related to the mouth and they sometimes show resistance – aggressive behavior, making the treatment approach complicated.

Moreover, several studies demonstrated the evidence of an association between periodontal disease and the onset/progression of dementia [14,15], although the specific mechanisms of this relationship need further investigation.

Another important aspect has to be considered: dementia patients management in assisted living facilities involves the collaboration of various specialists, in order to guarantee the diagnosis, treatment and follow-up of their several pathologies. However, among the various professional specialists, the dentist is often overlooked or, in many cases, not even considered [16].

### Rationale of the Study

The dentist has the task of evaluating the health condition of the oral cavity, which should be seen in the context of the patient’s general health. During the drawing of the dental treatment plan, medical problems of the patients (systemic diseases, pharmacological therapies, age, etc.) should be always considered. Dental approaches should be shaped according to both the oral and medical conditions of each subject.

The aging of the world population contributed to a rapid increase of dementia prevalence, giving it a predominant role in the context of systemic diseases. Elderly with dementia may be considered as *special needs* patients, towards which an ad hoc therapeutic approach should be used. This observational study had the purpose of evaluating the oral health status in nursing home resident’s elderly with dementia, analyzing the different possible causes that lead to a poor oral hygiene and suggesting the most appropriate specialist treatment approach.

## 2. Objectives

The primary objectives of this observational study were: (1) to evaluate the prevalence and the characteristics of oral pathology in a demented elderly population, (2) to check the association between the different degree of dementia and the oral health condition of each patient. (3) Furthermore, in the light of the results obtained, the adequacy of prevention and dental care in the area of residential care institutions will be evaluated and, by patient’s condition and needs, the most correct specialist dental approach will also be indicated.

## 3. Methods

For this observational study we used a cross-sectional design, in order to analyze oral health condition in elderly suffering from dementia at a specific point in time. This type of study is appropriate to investigate the prevalence of oral pathologies relationship between the cognitive deterioration and the oral hygiene of the population taken into consideration. The study was conducted according to the Declaration of Helsinki and independently approved and reviewed by the Institution’s Ethical Committee (#689/2014). A signed and written informed consent form was obtained from each patient, and if the patients would have been unable to do so, an informed written consent was obtained from their next of kin.

This study recruited two groups of elderly patients suffering from dementia, living in two different residential care institutions the first group had 18 subjects, of whom 14 were women and four were men and the second one consisted of 21 elderly, of whom 20 were women and one was a man, for a total of 39 patients, aged between 76 and 99 years (median age = 89.92 years old ± 5.3 SD). The severity of dementia of each included patient was performed using the Clinical Dementia Rating Scale [17], which provides a semi-structured interview with the patient and the caregiver/families, in order to rate the cognitive deterioration of the subject. The interview investigates six domains: memory, orientation, judgment/problem solving (cognitive domains), social activities, domestic activities/hobbies and personal care (functional domains). At the end of the interview, a score was assigned based on the impairment degree: 0 = normal, 0.5 = questionable/very mild, 1 = mild, 2 = moderate, 3 = severe, 4 = very severe, 5 = terminal. The sample of population included one subject with mild cognitive impairment (CDR = 1), nine subjects with moderate dementia (CDR = 2), fifteen elderly with severe dementia (CDR = 3), eleven patients with CDR = 4 and three patients with CDR = 5.

Inclusion criteria were as follows:Patients had to be diagnosed with firm dementia.Patients had to be resident in residential care institution.

Following subjects were excluded from the research:Patients with aggressive behavior, potentially dangerous for themselves and for the operator.

In order to evaluate the oral health condition of the included subjects, each patient in each group underwent a physical examination of the oral cavity, during which different clinical parameters were analyzed. To each parameter a score was assigned. Patient’s medical history was collected before proceeding with the oral examination (Table 1).


*Oral Examination*



Number of remaining teeth [18]:3 = total number of remaining teeth between 25 and 32 (complete or almost complete dentition), absence of prosthetic elements2 = partial edentulism, presence of at least 10 healthy/compromised/prosthetic dental elements1 = partial edentulism, presence of less than 10 healthy/compromised/prosthetic dental elements0 = complete edentulism and/or presence of retained rootsOral mucosa:1 = absence of lesions and/or new mucosal formations0 = presence of lesions and/or new mucosal formations, which may present the following characteristic [19]:
⮚acute/chronic onset ulcerative, vesciculo-bullous lesions⮚red, blue or red – purple plan/exophytic lesions, not removable white lesions, white and red, yellowish, pigmented lesions⮚Swelling on palate/tongue/oral floor/vestibular mucosa/gingivalPeriodontal tissues:3 = periodontal health [20] (code 0,1 or 2, basic periodontal examination, BPE):
⮚periodontal pockets ≤ 3 mm⮚negative or positive Bleeding on Probing Index (BoP)⮚absence or presence of limited quantities of plaque and calculus above and/or below the gum line and/or protruding filling materials
2 = generalized gingivitis with increased volume and gum redness, bleeding on probing due to plaque build-up, in the absence of clinical attachment loss or in presence of stable clinical attachment loss1 = chronic periodontitis of mild (CAL = 1–2 mm), moderate (CAL = 3–4 mm) or severe (CAL > 5 mm) degree [21]Bone crests [22]:
3 = absence of clinically detectable bone reabsorption (corresponding to class I or II according to Cawood and Howell)2 = moderate clinically detectable generalized bone reabsorption (corresponding to class III according to Cawood and Howell)1 = marked clinically detectable generalized bone reabsorption (corresponding to class IV or V according to Cawood and Howell)


After the oral examination the cooperation level of patients was also recorded:


*Level of Cooperation:*
4 = the patient is able to follow simple instructions (open your mouth, pull out your tongue, turn your head, report symptoms, etc.)3 = the patient is able to follow simple instructions for a limited period of time2 = the patient transposes but partially follows/doesn’t follow simple instructions, due to motor skills deterioration, abnormal and inappropriate reactions to the request1 = the patient transposes simple instructions but has no reaction0 = the patient does not transpose simple instructions and does not follow them



*Oral Examination:*
PossiblePartially possibleNot possible


The sum of the scores obtained in each parameter stated the oral health status. The maximum achievable scoring result is: 3 + 3 + 1 + 3 = 10. A score between 0 and 4 indicated severe oral health impairment, a score between 5 and 7 showed a moderate impairment, whereas a score between 8 and 10 stood for a mild impairment of the oral health.

### Statistical Methods

The prevalence and the characteristics of oral pathology in this sample of elderly with dementia were calculated using percentages and means. In order to evaluate the correlation coefficient between two independent quantitative variables, the degree of cognitive impairment and the oral health status of included subjects the Spearman’s Rho [23] test was used. The correlation index (*r*) was calculated taking into consideration the oral health values of patients with CDR between 2 and 5 (moderate – severe – very severe – terminal dementia). The values for *r* = –1 < r < 1 were considered significant, of which:r = –1 perfectly linear correlation (inverse relationship)−1 < r < 0 tendentially linear correlation (inverse relationship)r = 0 absence of linear correlation0 < r < 1 tendentially linear correlation (direct relationship)r = 1 perfectly linear correlation (direct relationship)

From the sample used for statistical analysis 10 patients were excluded, for whom it was impossible to perform the oral examination or part of it. However, the partial data of these patients have been reported within the research, in order to show how much the cooperation level could affect dental procedure.

## 4. Results

This research aimed at defining the prevalence of oral disease in a population of elderly patients with dementia and describing the possible presence of a linear correlation between the degree of dementia and the oral health status. The data collected during the oral examination have been reported in Table 2.

Regarding the prevalence of oral pathology, it emerged that 20.58% of the included patients (7/34) had mucosal lesions and/or new mucosal formations (in most cases undiagnosed and therefore untreated) and that, among these subjects, 8.82% (3/34) of the elderly presented potentially malignant lesions and/or new mucosal formations (ulcerated, bleeding lesions) (Figure 1).

The prevalence of periodontal disease was equal to 82.35% (25/34), of which just over half (58.82%: 20/34) was represented by clinical signs of gingivitis, while 8 patients out of 34 (23.52%) showed clinical signs of moderate – severe periodontal disease (Figure 2).

A marked clinically detectable reabsorption of bone crests was found in almost all patients (88.23%: 30/34), which may be associated with edentulism or the presence of severe periodontal disease.

24.13% (7/29) of patients, who underwent the oral examination, had totally edentulous maxillae and/or with retained roots, without prosthetic rehabilitations.

As for the health status of the oral cavity, scores from 3 to 8 were assigned. Only 34.48% (10/29) of the elderly achieved a score of 6, while more than half of the patients had a score less than 6 (55.17%:16/29), of which 17.24% (5/29) got a score of 5, 27.58% (8/29) a score of 4 and 10.34% (3/29) a score of 3. Only one patient with CDR = 0.5 (questionable cognitive impairment) got a score of 8, in the absence of periodontal disease, lesions and/or new mucosal formations and bone crests marked reabsorption.

The correlation index *r* showed the presence of a linear correlation (inverse relationship) between the degree of dementia and the state of health of the oral cavity of each patient. The test was performed taking into account the health status values of patients with CDR:
a)2–4–5, with the following results
*r* = –0.72031*p* < 0.001b)2–3–4–5, with the following results
*r* = –0.62982*p* < 0.001

According to the obtained results, the degree of poor oral health status tended to decrease as the degree of dementia increased, showing the presence of an inverse linear relationship between the two quantitative variables. The comparison between the average state of health in the different degrees of dementia confirmed the linear nature of the relationship between the two variables: in patients with CDR = 2 the average health status was 5.8, in patients with severe dementia - very severe (CDR = 3–4) was, respectively, equal to 4.9 and 4.4, while the terminal dementias (CDR = 5) reached an average equal to 3.5.

The deterioration of the oral health conditions could also be associated with the level of collaboration, which varied, in most cases, in relation to the dementia degree. Patients with CDR equal to 2 were able to follow simple instructions (open your mouth, pull out the tongue, turn your head, report symptoms) for a more or less limited period of time (33.3%: 3/9 with level of cooperation = 4; 66.67%: 6/9 with level of cooperation = 3) and none of them showed a level of cooperation of less than 3. In 20% (3/15) of the patients with severe dementia (CDR = 3) there was a level of cooperation equal to 2 (they transposed but partially followed/didn’t follow simple instructions, due to motor skills deterioration, abnormal and inappropriate reactions to the request), while 13.3% (2/15) transposed but had no reaction to requests. Out of 11 patients with very severe dementia, 3 (27.2%) did not transpose and therefore did not follow simple instructions (level of cooperation = 0). The totality of patients with terminal dementia did not exceed the level of cooperation = 0.

## 5. Discussion

In line with many authors [24,25], the results of this study showed that the higher the dementia degree is, the lower is the patient’s cooperation and lower the oral health status. Evidence of poor oral hygiene in elderly patients with dementia are reported in literature by several studies [10,26], which demonstrated a greater decline in the salivary glands function, an increase in the prevalence of edentulism (up to about 65%), of the accumulation of plaque and tartar, of periodontal sites with gingival bleeding and a greater incidence of coronal and root caries, compared to elderly subjects with normal cognitive functions. Moreover, most of the patients included in our study suffers from arterial hypertension, condition that could have an impact on oral health. Antihypertensive medications may cause oral dryness, that could contribute to several oral diseases. The study by Nonzee et al. [27] demonstrated that xerostomia, hyposalivation and increasing number of oral microbiota were more common in patients who underwent anti-hypertensive treatment. On the other hand, oral chronic inflammatory diseases, such as periodontal disease, may have a role in the onset of the mechanism for pro-hypertensive immune activation [28]. The medical–nursing teams of the residential care institutions included in this research stated that, in the management of their patients, dental care is, in most cases, overlooked and that the maintenance of the oral health of the subjects is carried out almost exclusively by family members. Literature reports a general reluctance by medical – nursing team to give priority to oral assistance, due to limited knowledge in the dental field and to psychological barriers regarding “working in the mouth of another person”. Furthermore, due to behavioral problems associated with dementia, patients often neglect oral hygiene, by refusing assistance from the facility staff. However, the management program of elderly patients suffering from dementia does not include any oral healthcare system, perpetuating the “oral abandonment” tendency [16,29]. Although doctors and nurses of residential care institution don’t have specific skills to guarantee basic dental management of patients, the figure of the dentist is not included in the medical team of these facilities. On the other hand, the possibility of performing specific dental therapies in patients with dementia is very low. Dental therapies always require certain compliance by the patient, in order to guarantee their maintenance over time; in fact, before proceeding with any type of treatment, the level of cooperation must be evaluated, which, if considered insufficient, makes the therapeutic act itself contraindicated. Moreover, the objectives of a dental intervention in the health of patients with dementia are limited by their high mortality. However, in order to guarantee even to the elderly with dementia the right to live free from pain or discomfort caused by the oral condition, the healthcare for residents in residential care institutions should include specific dental protocols, to maintain adequate parameters of oral health and, therefore, to improve the quality of the patient’s life [30].

Dentist and dental hygienist should cooperate, ensuring periodic checks and intervention to the patients suffering from dementia, according to individual needs. Dental care protocol for patients with dementia should include mechanical removal of plaque and food residue using toothbrushes, interproximal brushes, dental floss and fluoride-based toothpaste [16]. In patients with adequate levels of cooperation it is recommended to do rinse with chlorhexidine mouthwash at 0.2%, reducing bacterial load (*Lactobacillus, Streptococcus mutans*). For patients with lower levels of cooperation it is recommended to apply chlorhexidine gel using gauze. Furthermore, the use of salivary substitutes and limitations in sugar consumption, are appropriate preventive strategies against the development of caries. Finally, dental prosthesis hygiene should also be guaranteed: the dentist should periodically check the stability of the prosthesis, in order to avoid mucosal trauma. It is also important to thoroughly wash the prostheses, using gluconate chlorhexidine, benzalkonium chloride or sodium hypochlorite [31].

Several authors have demonstrated that residential care institutions nursing staff should be aware of the limited competence in dental field and that, in order to ensure adequate care, specific training courses regarding possible strategies for oral health maintenance in patients with special needs should be attended [29,32].

## 6. Strengths and Limitations of the Study

The limited number of the population sample and the lack of balance between male and female included patients represent a limitation of this study. On the other hand, a point of strength is the same method used to perform oral examination and the recording of all the medical information about clinical, behavioral and social condition of each included patient. Our study has several limitations. The population sample of this study may be not being extended enough and the number of the included male and female is not balanced. However, the same method was used for all patients to perform oral examination and all the medical information about clinical, behavioral and social condition of each patient were recorded.

## 7. Conclusions

The recent increase in life expectancy has led to an increase in the number of people living in old age, a phenomenon that will result in a growth in the number of elderly people with dementia by the middle of our century [33]. The deterioration of cognitive functions and that of motor and communication skills in dementia patients makes the correct execution of daily oral hygiene maneuvers difficult. The elderly with dementia may present behavioral disorders, with aggressive or opposing attitudes towards the medical–nursing staff of the residential care institutions, preventing the possibility of being assisted. A common condition among subjects suffering from dementia is, therefore, a poor oral hygiene. In the elderly with dementia there is a high prevalence of oral soft tissue pathologies, such as moderate–severe periodontal disease and gingivitis, which are rarely diagnosed and, therefore, rarely treated. Most patients with dementia have partially or totally edentulous arches, with marked bone reabsorption and the residual dental elements, if present, are often compromised (coronal/root caries, gingival recessions, periodontal disease). Retained roots may be source of painful symptoms, which is not always showed by dementia patients. The presence of a tendentially linear correlation between the degree of dementia and the oral health status has also been demonstrated: the greater the severity of dementia is, the lower is the level of oral health. Finally, one of the main obstacles that prevent adequate oral care for these patients in residential care institutions is the inadequate medical-nursing staff training on oral hygiene. It is therefore of primary importance to educate the nursing staff, in order to meet the special dental needs of patients with dementia.

## Figures and Tables

**Figure 1 ijerph-16-03763-f001:**
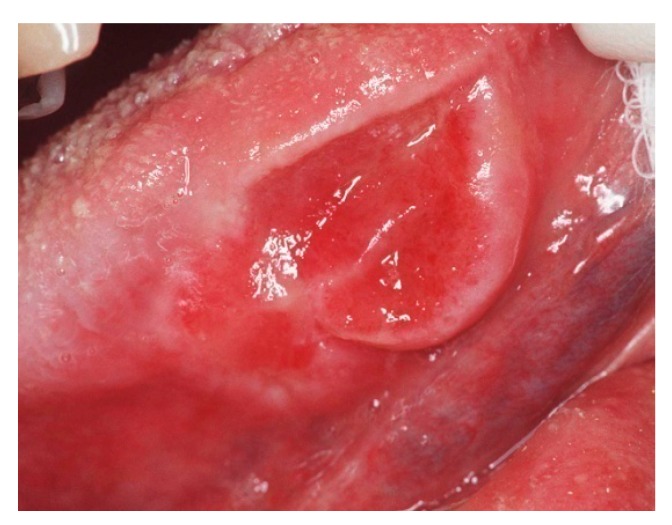
Ulcerative lesion of the border of the tongue: a biopsy was performed with histological examination that confirmed the suspicion of oral squamous cell carcinoma.

**Figure 2 ijerph-16-03763-f002:**
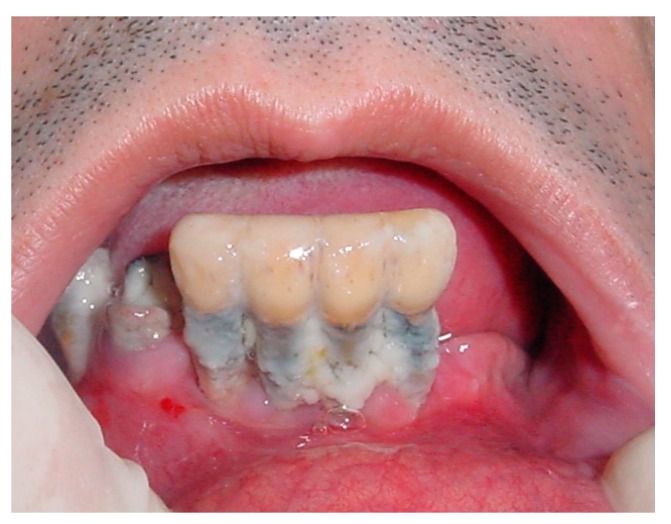
Severe periodontal disease and partial edentulism in subjects suffering from dementia.

**Table 1 ijerph-16-03763-t001:** Gender/Age, medical history of included patients.

Patients	Gender, Age	Medical History
Patient n. 1	woman, 88	- arterial hypertension - osteoporosis - anxiety syndrome - adequate nutritional status
Patient n. 2	man, 87	- arterial hypertension, chronic ischemic heart disease, percutaneous transluminal coronary angioplasty (PTCA) in 2002. - mesencephalic cerebral ischemia in chronic cerebral vascular disease. - chronic anemia - fairly good nutritional status
Patient n. 3	man, 98	- arterial hypertension - sleep disturbance - renal failure - adequate nutritional status
Patient n. 4	woman, 89	- arterial hypertension - chronic obstructive pulmonary disease (COPD) - mild-moderate renal failure - poor nutrition, soft diet
Patient n. 5	woman, 98	- arterial hypertension - hypothyroidism - cholelithiasis - MGUS - liver disease - right mastectomy - moderate malnutrition, food supplement
Patient n. 6	woman, 93	- arterial hypertension, atrial fibrillation, hypertensive heart disease, NSTEMI, currently in NOAC’s - dyslipidaemia - diabetes mellitus - right mastectomy - nutritional status in decrease
Patient n. 7	woman, 92	- arterial hypertension, atrial fibrillation in oral anticoagulant therapy - diabetes mellitus 2 - anemia - lobar pneumonia - moderate malnutrition
Patient n. 8	man, 85	- cholelithiasis - moderate neurogenic dysphagia - fairly good nutritional status
Patient n. 9	woman, 88	- arterial hypertension - chronic obstructive pulmonary disease (COPD) - osteoporosis - moderate malnutrition
Patient n. 10	woman, 90	- invasive ductal carcinoma (2005) - carotid angiosclerosis - poor nutritional status
Patient n. 11	woman, 80	- arterial hypertension - subacute bilateral subdural hematoma - adequate nutritional status
Patient n. 12	woman, 92	- arterial hypertension, atrial fibrillation in oral anticoagulant therapy - right hemicolectomy (1996) - adequate nutritional status
Patient n. 13	woman, 93	- diabetes mellitus 1 - adequate nutritional status
Patient n. 14	woman, 89	- arterial hypertension, atrial fibrillation (placement of pacemaker) - diffused arthrosis - adequate nutritional status
Patient n. 15	man, 92	- arterial hypertension, chronic atrial fibrillation in oral anticoagulant therapy - dyslipidaemia - diabetes mellitus 2 - chronic renal failure - hypothyroidism - nutritional status: obesity
Patient n. 16	woman, 81	- diabetes mellitus 2 - bedridden with PEG
Patient n. 17	woman, 82	- chronic atrial fibrillation in oral anticoagulant therapy - depression - fairly good nutritional status
Patient n. 18	woman, 95	- hypertensive heart disease, venous thrombosis (left lower limb, 2018) - hysterectomy - seborrheic dermatitis (face) - chronic blepharitis - moderate malnutrition
Patient n. 19	woman, 96	- arterial hypertension - osteoporosis - mammary carcinoma - fairly good nutritional status
Patient n. 20	woman, 90	- arterial hypertension, atrial fibrillation in oral anticoagulant therapy - osteoarthritis - adequate nutritional status
Patient n. 21	woman, 88	- arterial hypertension, chronic ischemic heart disease with acute myocardial infarction - chronic obstructive pulmonary disease (COPD) - diabetes mellitus 2 - osteoporosis - adequate nutritional status
Patient n. 22	woman, 91	- arterial hypertension, venous insufficiency, von Willebrand’s disease - chronic obstructive pulmonary disease (COPD) - moderate malnutrition
Patient n. 23	man, 76	- arterial hypertension, atrial fibrillation in oral anticoagulant therapy, bacterial endocarditis with mitral insufficiency and valve replacement, coronary artery bypass in oral anticoagulant therapy - chronic cerebral vascular disease - diabetes mellitus 2 - chronic renal failure - adequate nutritional status
Patient n. 24	woman, 90	- thrombophlebitis left lower limb - facial meningioma - diffused arthritis, gait disturbances - adequate nutritional status
Patient n. 25	woman, 79	- hypothyroidism - lumbosacralspondylosis - adequate nutritional status
Patient n. 26	woman, 93	- arterial hypertension - cerebral ischemic vascular disease - mammary carcinoma - osteoporosis - adequate nutritional status
Patient n. 27	woman, 96	- arterial hypertension, atrial fibrillation in oral anticoagulant therapy, tricuspid insufficiency mild/moderate, pacemaker - previous gonarthrosis - bedridden, with nasogastric tube
Patient n. 28	woman, 91	- arterial hypertension - osteoporosis, gait disturbances - adequate nutritional status
Patient n. 29	woman, 91	- arterial hypertension - colon diverticulosis - osteoporosis, gait disturbances - adequate nutritional status
Patient n. 30	woman, 94	- arterial hypertension - cerebral vascular disease - visual disturbance - anorexia
Patient n. 31	woman, 92	- arterial hypertension - left-sided ischemic stroke (2015) - pneumonia - kidney stones - diffused arthrosis - PEG, then removed, currently moderate malnutrition
Patient n. 32	woman, 85	- chronic cerebral vascular disease, ischemic event with a right hemiplegia outcome - gait disturbances - fairly good nutritional status
Patient n. 33	woman, 91	- arterial hypertension - ischemic vascular disease - diffused arthrosis, gait disturbances - adequate nutritional status
Patient n. 34	woman, 84	- arterial hypertension - uterus-vaginal prolapsed, rectocele - osteoporosis, gait disturbances - adequate nutritional status
Patient n. 35	woman, 99	- arterial hypertension - chronic cerebral vascular disease with hemorrhagic stroke (2012) - osteoporosis, diffused arthrosis, gait disturbances - sleep disorders - fairly good nutritional status
Patient n. 36	woman, 90	- arterial hypertension, chronic venous insufficiency - chronic cerebral vascular disease - chronic peripheral neuropathy in diabetes mellitus 2 - glaucoma - osteoporosis, diffused arthrosis - chronic renal failure - adequate nutritional status
Patient n. 37	woman, 94	- arterial hypertension, aortic stenosis - cerebral ischemic vascular disease - chronic obstructive pulmonary disease (COPD) - gait disturbances - adequate nutritional status
Patient n. 38	woman, 93	- low hearing, ear anomalies - adequate nutritional status
Patient n. 39	woman, 92	- chronic heart disease with mild-moderate valvulopathy - hypothyroidism - chronic liver disease with HCV infection - sensory polyneuropathy (lower limbs) - osteoporosis, Dupuytren syndrome (right hand), gait disturbances - adequate nutritional status

MGUS: Monoclonal Gammopathy of Undetermined Significance; NOACs: Non-vitamin K oral anticoagulants; NSTEMI: Non-ST elevation myocardial infarction; PEG: percutaneous endoscopic gastrostomy.

**Table 2 ijerph-16-03763-t002:** CDR, Level of Cooperation, Oral Health status.

Patients	CDR	Level of Cooperation	Oral Examination
Patient n. 1	2	4 = the patient is able to follow simple instructions (open your mouth, pull out your tongue, turn your head, report symptoms, etc.)	Possible Number of remaining teeth = 1 Oral mucosa = 1 Periodontal tissues = 3 Bone crests = 1 **Oral health status = 6**
Patient n. 2	3	3 = the patient is able to follow simple instructions for a limited period of time (open your mouth, pull out your tongue, turn your head, report symptoms, etc.)	Partially possible Number of remaining teeth = 0 Oral mucosa = 1 Periodontal tissues = 2 Bone crests = 1 **Oral health status = 4**
Patient n. 3	2–3	3 = the patient is able to follow simple instructions for a limited period of time (open your mouth, pull out your tongue, turn your head, report symptoms, etc.)	Partially possible Number of remaining teeth = 0 Oral mucosa = 1 Periodontal tissues = 3 Bone crests = 1 **Oral health status = 5**
Patient n. 4	4	3 = the patient is able to follow simple instructions for a limited period of time (open your mouth, pull out your tongue, turn your head, report symptoms, etc.)	Partially possible Number of remaining teeth = 0 Oral mucosa = 1 Periodontal tissues = 2 Bone crests = 1 **Oral health status = 4**
Patient n. 5	3	4 = the patient is able to follow simple instructions (open your mouth, pull out your tongue, turn your head, report symptoms, etc.)	Possible Number of remaining teeth = 1 Oral mucosa = 1 Periodontal tissues = 2 Bone crests = 1 **Oral health status = 5**
Patient n. 6	3	2 = the patient transposes but partially follows/doesn’t follow simple instructions, due to motor skills deterioration, abnormal and inappropriate reactions to the request	Impossible
Patient n. 7	3	3 = the patient is able to follow simple instructions for a limited period of time (open your mouth, pull out your tongue, turn your head, report symptoms, etc.)	Partially possible Number of remaining teeth = undetectable Oral mucosa = 1 Periodontal tissues = 2 Bone crests = 2 **Oral health status = undetectable**
Patient n. 8	4	0 = the patient does not transpose simple instructions and does not follow them	Partially possible Number of remaining teeth = undetectable Oral mucosa = 1 Periodontal tissues = 3 Bone crests = 2 **Oral health status = undetectable**
Patient n. 9	4	3 = the patient is able to follow simple instructions for a limited period of time (open your mouth, pull out your tongue, turn your head, report symptoms, etc.)	Partially possible Number of remaining teeth = 2 Oral mucosa = 1 Periodontal tissues = 2 Bone crests = 1 **Oral health status = 6**
Patient n. 10	4	2 = the patient transposes but partially follows/doesn’t follow simple instructions, due to motor skills deterioration, abnormal and inappropriate reactions to the request	Partially Possible Number of remaining teeth = 0 Oral mucosa = 1 Periodontal tissues = 2 Bone crests = 1 **Oral health status = 4**
Patient n. 11	4	0 = the patient does not transpose simple instructions and does not follow them	Impossible
Patient n. 12	0,5	4 = the patient is able to follow simple instructions (open your mouth, pull out your tongue, turn your head, report symptoms, etc.)	Possible Number of remaining teeth = 2 Oral mucosa = 1 Periodontal tissues = 3 Bone crests = 2 **Oral health status = 8**
Patient n. 13	3	3 = the patient is able to follow simple instructions for a limited period of time (open your mouth, pull out your tongue, turn your head, report symptoms, etc.)	Partially possible Number of remaining teeth = undetectable Oral mucosa = 1 Periodontal tissues = 2 Bone crests = 1 **Oral health status = undetectable**
Patient n. 14	2	3 = the patient is able to follow simple instructions for a limited period of time (open your mouth, pull out your tongue, turn your head, report symptoms, etc.)	Partially possible Number of remaining teeth = 2 Oral mucosa= 1 Periodontal tissues = 1 Bone crests = 2 **Oral health status = 6**
Patient n. 15	3	2 = the patient transposes but partially follows/doesn’t follow simple instructions, due to motor skills deterioration, abnormal and inappropriate reactions to the request	Impossible
Patient n. 16	5	0 = the patient does not transpose simple instructions and does not follow them	Partially possible Number of remaining teeth = undetectable Oral mucosa = 0 Periodontal tissues = 2 Bone crests = 1 **Oral health status = undetectable**
Patient n. 17	5	0 = the patient does not transpose simple instructions and does not follow them	Partially possible Number of remaining teeth = 0 Oral mucosa = 1 Periodontal tissues = 2 Bone crests = 1 **Oral health status = 4**
Patient n. 18	3	2 = the patient transposes but partially follows/doesn’t follow simple instructions, due to motor skills deterioration, abnormal and inappropriate reactions to the request	Partially possible Number of remaining teeth = undetectable Oral mucosa = 1 Periodontal tissues = 2 Bone crests = 1 **Oral health status = undetectable**

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
