# Peer review of "Aging and Oral Care: An Observational Study of Characteristics and Prevalence of Oral Diseases in an Italian Cohort"

_ijerph, 2019, doi:10.3390/ijerph16193763_

Round 1

Reviewer 1 Report

Line 139: The diagnosis of dementia of each...... May be: The severity of dementia? Diagnostic criteria for dementia: DSM-4 or DSM-5. Line 138: middle age - 89.67±SD Line 228: he  - the Figure 2: this figure has colourful borders while other figures do not Line 260, 263: if p is 0.0005 or 0.00025, then usually it would be written as p<0.001. Line 264: "the oral health status tended to decrease..". May be: the degree of poor oral health status? Line 273: open your mouth" - open your mouth. Table 3.1: patient n2 - chronicanemia - chronic anemia. Table 3.1: patient n5 - "mediocre nutritional status". Very strange definition. May be: "moderate malnutrition"? Table 3.1: patient n10 - invasive ductal carcinoma, mammary carcinoma. It it the same diagnosis Table 3.1: patient n11 - sub acute - subacute. Table 3.1: patient n12 - Diffused - diffused. Table 3.1: patient n13 - emicolectomy - hemicolecectomy. Table 3.1: patient n17 - Bed ridden with PEG - bedridden, with PEG. Table 3.1: patient n19 - Chronich lepharitis - chronic blepharitis, mediocre nutritional status? Line 286:myocardical infarction - myocardial infarction. Table 3.1 and Table 3.2 must be Table 3 and and Table 4. Table 3.2: patient n9 - bedridden with .... - bedridden, with ... Table 3.2: patient n17 - with hemorrhagic ictus? Personally, I do not think there is a point in showing previous conditions - it is unlikely that they could be linked with oral status (except nutritional status), and such linkage is nor even considered in this study. Both patient groups should be joined to make one group because data has not been compared between groups. This is important because your aim is to study the link between oral health status and dementia severity. Line 292:"lower is also the oral health status"?  The conclusion is incorrect, as data has not been compared with those who do not have dementia.

Author Response

Milan 25/09/2019         

Dear Reviewer,

                  many thanks for the insightful comments and suggestions of the referees. We have made corresponding revision according to their advice. Words in red are the changes we have made in the text. The language of the manuscript has also been extensively revised by a professional MDPI English language science editing service and all authors of this article have seen and approved the changes.

The revisions are as follows:

Line 148: “The severity of dementia” instead of “The diagnosis of dementia”

Line 147: ± 5.3 (SD)

Line 245: “the” instead of “he”

Figure 1-2: we deleted colorful borders in both figures

Line 279-282: p < 0.001 instead of p = 0.0005/0.00025

Line 283: “the degree of poor oral health status tended to decrease” instead of “the oral health status tended to decrease”

Line 292: open your mouth instead of open your mouth”

Tables: both groups were joined in the same group.

Table 1: CDR, Level of Cooperation, Oral Health status Table 2: Gender/Age, medical history of included patients

Table 2:

patient n. 2 à chronic anemia instead of chornicanemia patient n. 5-7-9 à “moderate malnutrition” instead of “mediocre nutritional status” patient n. 10 à we deleted mammary carcinoma patient n. 11 à subacute instead of sub acute patient n. 12 à hemicolectomy instead of emicolectomy patient n. 16 à bedridden instead of bed ridden patient n. 18 à chronic blepharitis instead of chronich lepharitis patient n. 22-31 à “moderate malnutrition” instead of “mediocre nutritional status” patient n. 27 à “bedridden, with” instead of ”bedridden with” patient n. 35 à “with hemorrhagic stroke” instead of “with hemorrhagic ictus”

Line 358: “myocardial infarction” instead of “myocardical infarction”

Line 245: “This research aimed at defining the prevalence of the oral disease in a population of elderly patients with dementia and describing the possible presence of a linear correlation between the degree of dementia and the oral health status”. As explained in the first part of the “Results”, our study had the aim to compare the different degree of dementia (following CDR) and the related oral health status, not considering people without cognitive impairment.

Regarding medical history: “For this reason, the treatment approach in dentistry should always include medical history, in order to obtain the greatest possible number of information on the patient's state of health. Collecting medical history has the objective of defining whether or not it is appropriate to proceed with therapy, to verify whether the patient is able to sustain the treatment and if some drugs are contraindicated. Jainkittivonget al. highlighted that medical consultations could reduce the medical risk associated with dental procedures and unnecessary antibiotic prophylaxis. Therefore, for many dental patients, good communication between dentists and physicians is essential for inappropriate care. Unfortunately, the importance of medical history in oral care is often underestimated and it is consequently often neglected”. One of the main concepts of our review is that the dentist should always collect the patient’s medical history (and not only his dental history), in order to choose the most appropriate therapy. Moreover, several systemic conditions, such as arterial hypertension, may have an impact on oral health, a topic that has been discussed in the “Discussion”.

Thank you for receiving our manuscript and considering it for publication.

We appreciate your time and look forward to your response.

Yours sincerely,

Dorina Lauritano

Reviewer 2 Report

EDITS (Number refers to line number)

51 should instead of shall

67 delete anamnesis as it is not the patient’s recollection of their medical history but medical “protocol”

72 delete inadequate and replace it with appropriate

Anamnesis is defined as “a patient's account of their medical history” I am not sure that this is the thesis of this paper. I would eliminate this sentence completely. Has no bearing on the paper.

78 “…prefiguring of alarming dimensions:” replace with “projected to become overwhelming:”

99 oppositive replace with resistance

106 overshadowed replace with overlooked

109 inserted replace with seen

109 conception replace with drawing

138 (Middle age=89,67 years old) replace with (Median=89.67 years old)

218 iners replace with inverse

255 delete tendentially

292 delete ‘tendentially’  and ‘is also’ to read ‘,the lower is the patient’s cooperation and lower the oral health status.”

COMMENT

A primary comorbidity is arterial hypertension which must play a role in some of the oral health complications that were observed. A comment in the conclusion needs to address this. A few sentences to acknowledge this finding and to also suggest that by itself hypertension promotes oral decay.

The main feature of this paper is suggestions for care treatment. This can make an impact in how assisted living place and nursing homes deal with oral care of their patients. 

Author Response

Milan 25/09/2019         

Dear Reviewer,

                  Many thanks for the insightful comments and suggestions of the referees. We have made corresponding revision according to their advice. Words in red are the changes we have made in the text. The language of the manuscript has also been extensively revised by a professional MDPI English language science editing service and all authors of this article have seen and approved the changes.

We have corrected our paper as follows:

Line 52: “it should be extended...” instead of “it shall be extended...”

Line 68: we deleted anamnesis and replaced it with “medical history

Line 73: “inappropriate care”

Line 80: “the impact of the phenomenon is projected to become overwhelming” instead of “the impact of the phenomenon is prefiguring of alarming dimensions”

Line 108: “resistance– aggressive behavior” instead of “ oppositive-aggressive behaviour”

Line 115: “the dentist is often overlooked” instead of “the dentist is often overshadowed”

Line 118: “which should be seen in the context of...” instead of “ which should be inserted in the context of...”

Line 118: “During the drawing of the dental treatment plan...” instead of “During the conception of the dental treatment plan...”

Line 147: “Median age = 89.67 years old” instead of “Middle age = 89.67 years old”

Line 235: Line 292: “inverse relationship”

Line 272-361: we deleted tendentially. “the lower is the patient’s cooperation and lower the oral health status” instead of “ the lower is the patient’s cooperation and lower is also the oral health status”

We added a few sentences with references regarding the relationship between hypertension and oral health. Line 366: ”Moreover, most of the patients included in our study suffers from arterial hypertension, a condition that could have an impact on oral health. Antihypertensive medications may cause oral dryness, that could contribute to several oral diseases. The study by Nonzee et al.[1] demonstrated that xerostomia, hyposalivation and increasing number of oral microbiota were more common in patients who underwent anti-hypertensive treatment. On the other hand, oral chronic inflammatory diseases, such as periodontal disease, may have a role in the onset of the mechanism for pre-hypertensive immune activation[2].”

Thank you for receiving our manuscript and considering it for publication.

We appreciate your time and look forward to your response.

Yours sincerely,

Dorina Lauritano

[1] Nonzee V, Manopatanakul S, Khovidhunkit SO. Xerostomia, hyposalivation and oral microbiota in patients using antihypertensive medications. J Med Assoc Thai 2012 Jan;95(1):96-104.

[2] Marta Czesnikiewicz-Guzik, Grzegorz Osmenda3, Mateusz Siedlinski, Richard Nosalski, Piotr Pelka, Daniel Nowakowski, Grzegorz Wilk, Tomasz P. Mikolajczyk , Agata Schramm-Luc , Aneta Furtak, Pawel Matusik, Joanna Koziol, Miroslaw Drozdz, Eva Munoz-Aguilera, Maciej Tomaszewski , Evangelos Evangelou, Mark Caulfield, Tomasz Grodzicki, Francesco D’Aiuto, and Tomasz J. Guzik. Causal association between periodontitis and hypertension: evidence from Mendelian randomization and a randomized controlled trial of non-surgical periodontal therapy. Eur Heart J. 2019 Sep 1. pii: ehz646.

Round 2

Reviewer 1 Report

Thank you for your correction.

Some notes:

Table 2: patient n12 - right hemicolecectomy (1996) - hemicolectomy. Sorry, it was my mistake..

Table 2: patient n16 - Bedridden - bedridden.

Table 2: patient n18 - Chronic - chronic.

I think, that the conclusion "Elderly with dementia have a higher prevalence of oral pathologies compared to those with normal cognitive functions" is incorrect, as data has not been compared with those who do not have dementia. The one of objectives of your study was: "..to check the association between the different degree of dementia and the oral health condition of each patient".

Author Response

Milan 02/10/2019         

            Dear Reviewer,

            many thanks for the insightful comments and suggestions of the referees. We have made corresponding revisions according to their advice. Words in red are the changes we have made in the text. The language of the manuscript has also been extensively revised by a professional MDPI English language science editing service and all authors of this article have seen and approved the changes.

We have modified our study as follows:

Table 2 à “hemicolectomy” instead of “hemicolecectomy” “bedridden” instead of “Bedridden” “chronic” instead of “Chronic”

We have also corrected some sentences of the “Conclusions”

Line 43 à “Several factors contribute to poor oral health in the elderly suffering from dementia: cognitive functions deterioration, behavioral disorders and inadequate medical-staff nursing training on oral hygiene. This study also demonstrated that the lower the dementia degree is the lower tends to be the oral health status. In order to guarantee complete assistance to these patients, residential care institutions should include in their healthcare program-specific dental protocols”.

Line 441à ”A common condition among subjects suffering from dementia is, therefore, poor oral hygiene. In the elderly with dementia, there is a high prevalence of oral soft tissue pathologies, such as moderate-severe periodontal disease and gingivitis, which are rarely diagnosed and, therefore, rarely treated”.

Thank you for receiving our manuscript and considering it for publication.

We appreciate your time and look forward to your response.

Yours sincerely,

Dorina Lauritano
